# Efficacy of a Modified-Live Virus Combination Vaccine (CDV, CAV, CPV, CPiV), Canigen^TM^ DHPPi, in Puppies Vaccinated at Six Weeks of Age

**DOI:** 10.3390/v17121607

**Published:** 2025-12-12

**Authors:** Sofia Loukeri, Fabien Senseby, Elodie Benizeau, Joelle Cronier, Sylvie Gueguen

**Affiliations:** Virbac, 13ème rue, LID, 06511 Carros, France; fabien.senseby.external@virbac.com (F.S.); elodie.benizeau@virbac.com (E.B.); joelle.cronier@virbac.com (J.C.); sylvie.gueguen@virbac.com (S.G.)

**Keywords:** vaccine, vaccination, dog, puppy, efficacy, distemper, hepatitis, parvovirus, parainfluenza, challenge, seroconversion

## Abstract

During their early life, puppies are protected against infectious agents with the presence of maternal derived antibodies (MDA). Vaccination is recommended to start as soon as the levels of MDA begin to wane to ensure that the puppy’s immune system can respond effectively to the vaccines and develop active immunity against diseases. Two studies were designed to assess the efficacy of the Canigen^TM^ DHPPi vaccine in puppies from 6 weeks of age. The studies comprised two parts: the efficacy assessment of the Canine Parainfluenza Virus (CPiV) vaccine component against a virulent challenge with CPiV (Experiment A) and an immunogenicity assessment of Canine Distemper Virus (CDV), Canine Adenovirus (CAV-2), and Canine Parvovirus (CPV-2) vaccine components (Experiment B). In Experiment A, twelve puppies were immunized with two injections of Canigen^TM^ DHPPi (at minimum titer) two weeks apart and twelve control puppies received the vaccine diluent. All animals were challenged with a virulent, heterologous CPiV strain two weeks after the second vaccination. Vaccinated puppies exhibited a significant reduction in nasal viral shedding compared to the control group. Clinical signs of respiratory disease were mild and transient in both groups. In Experiment B, six puppies were immunized with two injections of Canigen^TM^ DHPPi (at minimum titer) two weeks apart. A follow-up of the seroneutralizing antibodies titers against the CDV, CAV-2 and CPV-2 vaccine components was performed in order to assess the efficacy on the serological response basis. After the first vaccine injection, all the puppies seroconverted and presented seroneutralizing antibody titers reaching a protective thresholds against CDV (≥10^0.82^), CAV-2 and CAV-1 (≥10^0.82^), CPV-2 and CPV-2c (≥10^1.8^). After the second vaccine injection, a more robust immune response was observed and the seroneutralizing antibody titers remained high until 4 weeks post vaccination for those three vaccine components. In both experiments (A and B), all vaccinated animals remained in good health, with no adverse reactions recorded during the vaccination phase. As a conclusion, the efficacy of the Canigen^TM^ DHPPi vaccine was demonstrated when administered in dogs from 6 weeks of age. These results fully support the interest of an early vaccination in such young puppies followed by the recommended vaccination scheme.

## 1. Introduction

Canine infectious diseases, such as Canine Distemper Virus (CDV), Canine Adenovirus (CAV), Canine Parvovirus (CPV), and Canine Parainfluenza Virus (CPiV), are highly contagious and pose significant health threats to dogs in some cases with high mortality, particularly in young puppies [1,2]. Vaccination remains an important part of comprehensive preventive health care [3], and early and effective vaccination strategies are crucial for establishing protective immunity and mitigating the risk of severe disease. As a general rule, puppy vaccination courses should start as soon as the levels of maternal derived antibodies (MDA) begin to wane to ensure that the puppy’s immune system can respond effectively to the vaccines and develop active immunity against diseases. However, the optimal time of the initial vaccination for each individual puppy is unknown, and therefore early and repeated vaccinations can ensure the mounting of an effective immune response even for puppies with persistent MDA levels until 16 weeks of age or older [3]. This study aimed to evaluate the efficacy of the commercially available Canigen^TM^ DHPPi vaccine, a combination vaccine designed to protect against these key viral agents in puppies as young as six weeks of age. Two independent studies were conducted: a challenge study to assess the efficacy of the CPiV component based on viral shedding and clinical observation following a heterologous CPiV challenge, and an immunogenicity study assessing the ability of the vaccine to induce protective seroneutralizing antibody responses against CDV, CAV, and CPV.

## 2. Materials and Methods

Two experimental studies were conducted for the efficacy assessment of Canigen^TM^ DHPPi. The first (Experiment A) was a challenge study for the assessment of the Canine parainfluenza component and the second study (Experiment B) was an immunogenicity study to assess the ability of Canigen^TM^ DHPPi to induce protective levels of seroneutralizing antibodies against Canine Distemper Virus (CDV), Canine Adenovirus (CAV-2 and CAV-1) and Canine Parvovirus (CPV-2 and CPV-2c).

The challenge study was designed as an unblinded, randomized controlled trial, and the serological study was a single-cohort non-blinded trial.

Both trials were conducted in compliance with the recommendations issued in the European Pharmacopoeia and current legislations [4,5,6]. The animals were housed and cared for in accordance with the animal welfare standards described in the European Directive 2010/63/EU for the protection of laboratory animals [7].

### 2.1. Animals and Housing

All dogs were housed in controlled conditions with appropriate veterinary care and monitoring throughout the studies.

#### 2.1.1. Challenge Study (Experiment A)

As demonstrated in the literature, the only way to confirm true protection is to directly expose vaccinated animals to a virulent form of the virus (a challenge study) and observe the prevention or reduction in clinical signs and viral shedding. Antibody level (seroneutralization titer) is not reliably indicative of protection against Canine Parainfluenza, given that efficacy relies heavily on local, mucosal immunity in the respiratory tract, which is poorly correlated with serum antibody titers [8]. Twenty-four conventional 6 week old Beagle puppies (9 males and 15 females) were randomly assigned to two groups of 12 animals each. They were born from dams regularly vaccinated against CDV, CAV-2, CPV and CPiV. All puppies were tested for seroneutralizing antibodies against CDV, CAV-2, CPV and CPiV at 3 to 4 weeks of age.

The puppies were between 5.5 and 6.5 weeks old at the beginning of the study. During the study, the dogs were relocated once from the vaccination site to the virus challenge site at day 20.

#### 2.1.2. Serological Study (Experiment B)

Six Specific Pathogen Free (SPF) Beagle puppies (5 males and 1 female), MDA free, were included in the study. The puppies were 6 weeks old at the beginning of the study (between 39 and 43 days) and were born from dams that were not vaccinated against CDV, CAV-2, CPV and CPiV. They were all confirmed seronegative against all the vaccine components before vaccination.

### 2.2. Vaccination Schedule and Vaccine

The puppies of both studies apart from the control group were immunized with two subcutaneous injections of 1 mL of the commercial vaccine Canigen^TM^ DHPPi (Virbac, Carros, France) adjusted at a minimum titer, administered at a two-week interval (D0 and D14). The vaccination was administered at the interscapular region. The control group of the challenge study did not receive vaccination.

### 2.3. Efficacy Assessment

The efficacy of the Canine parainfluenza (CPiV) component was assessed through a virulent heterologous CPiV challenge performed two weeks after the second vaccination (T0). The efficacy of the Canine Distemper (CDV), Canine Adenovirus (CAV-1 and CAV-2) and Parvovirus (CPV-2 and CPV-2c) components was evaluated on the basis of the humoral immune response, two weeks after the first vaccination and four weeks after the second vaccination.

#### 2.3.1. Virus Challenge (Experiment A)

All animals underwent a challenge with a heterologous virulent CPiV strain, which was provided by the United States Department of Agriculture (USDA). After propagation on cells and inoculation on dogs, nasal and tracheal mucosa containing CPiV were recovered, processed and amplified on PRDK cells to establish the challenge strain stored in liquid nitrogen. The final challenge inoculum was prepared by diluting the strain to 5 × 10^3^ CCID_50_/mL. The challenge dose was validated in dose titration studies demonstrating that 4.10^5^ CCID_50_/dog consistently induced both viral shedding and respiratory clinical signs in 12–13 week old dogs. This dose was therefore selected as the minimum effective challenge dose. On T0, all dogs received 1 mL of the challenge strain via the intranasal route (0.5 mL per nostril with atomizer) and 1 mL via the intra-tracheal route, corresponding to 10^4^ CCID_50_ per dog.

The efficacy of the CPiV component was evaluated on the basis of the clinical signs and viral nasal shedding post inoculation.

The health status of all animals was monitored with daily clinical observation, complete clinical examinations and body weight recording. Clinical assessments were divided into two phases:Vaccination phase (D0–D28): Weekly clinical examinations, including rectal temperature measurement and body weight recording, were performed by a trained veterinarian or technician.Challenge phase (T0–T14): Daily clinical examinations were conducted, with particular attention to clinical signs indicative of respiratory disease, including nasal discharge, sneezing, coughing, dyspnea, throat redness and dehydration. Body weight was recorded weekly in a fasted state.

#### 2.3.2. Serological Response (Experiment B)

A monitoring of seroneutreutralizing antibody titers was performed throughout the study period for the efficacy assessment of the Canine Distemper component (CDV), Canine adenovirus component (CAV-2 and CAV-1) and parvovirus component (CPV2 and CPV-2c). Blood samples for serological analyses were collected before each vaccination and then on a weekly basis (D28, D35, D42) until the end of the study. In addition to the serological follow-up, a complete clinical examination, including rectal temperature monitoring and weighing, was carried out once a week in all dogs, from D0 to D42.

### 2.4. Sample Collection, Processing and Analysis

Blood and nasal swab samples were collected for serological and virological analyses.

#### 2.4.1. Challenge Study

Blood samples: Collected on uncoated tubes before each vaccination (D0 and D14), before challenge (T0), and at the end of the challenge phase (T14). Serum samples were centrifuged, aliquoted, and stored at ≤−15 °C until serological analysis.Nasal swabs: Collected from each nostril (1 swab per nostril), placed in PBS solution, stored under refrigerated conditions until processing, and then stored at ≤−70 °C until the quantification of CPiV.

#### 2.4.2. Serological Study

Blood samples were collected before each vaccination (D0 and D14), and then on a weekly basis (D28, D35, D42) until the end of the study. Serum samples were centrifuged, aliquoted, and stored at ≤−15 °C until serological analysis.

#### 2.4.3. Seroneutralization Techniques

Seroneutralization assays were conducted to evaluate antibody responses against canine distemper virus (CDV), canine adenovirus type 1 and type 2 (CAV-1 and CAV-2), canine parainfluenza virus (CPiV), and canine parvovirus serotypes 2 and 2c (CPV-2 and CPV-2c). Following a 6- to 7-day incubation period, the cytopathogenic effect (CPE) of residual virus particles not seroneutralized by antibodies was detected using immunofluorescence (IF) staining with virus-specific antibodies.

The designated positivity thresholds are based on internal studies validating the correlation of protection between seroneutralizing antibodies and protection against clinical disease and mortality following a virulent challenge for the CDV, CPV and CAV-2 vaccine components. The study reports were submitted to European regulatory bodies such as the European Medicines Agency (EMA).

#### 2.4.4. Quantification of Canine Parainfluenza Virus (CPiV) by RT-qPCR 

CPiV RNA was extracted from the nasal swabs, aliquoted, and stored at ≤−70 °C until quantification by RT-qPCR. RNA extraction was performed, which employs magnetic particle technology for automated nucleic acid purification from 200 µL of nasal swab samples supplemented with exogenous Xeno RNA.

Quantitative RT-PCR was performed by simplex RT-PCR, with fluorescence measured on the FAM channel. The limit of quantification (LOQ) was 186 CPiV copies/µL of nucleic acids (4.9 Log_10_/mL of nasal swabs), while the limit of detection (LOD) was 26 CPiV copies/µL (4.1 Log_10_/mL of nasal swabs).

### 2.5. Statistical Analysis

For the challenge study, the viral excretion (log_10_ copies/mL of swab) between T2 and T14 was analyzed using a general linear mixed model (Fixed effects: group, time and time × group interaction and animal as the subject term). The covariance matrix structure minimizing the Akaike Information Criterion (AIC) was selected. The Kenward–Roger method was applied to estimate degrees of freedom for testing fixed effects. All statistical analyses were conducted using SAS 9.4^®^, with a significance threshold of 5% (two-sided), except for normality testing, which used a 1% threshold.

## 3. Results

### 3.1. Experiment A (Challenge Study)

#### 3.1.1. Clinical Examination: Clinical Signs, Body Weight and Rectal Temperature

Vaccination Phase (D0–D28):

Throughout the vaccination phase (D0–D28), all dogs remained in good general health, maintaining normal physiological rectal temperatures (37.5–39.5 °C) and exhibiting the expected weight gain for puppies of such age. No adverse reactions to the vaccination were observed. A transient growth slowdown and minor, non-clinically relevant clinical signs in two dogs (one vaccinated, one control) were noted after relocation to the challenge site (D21). These effects were attributed to environmental changes, and all animals were fully adapted and healthy by D28 (T0).
Challenge Phase (T0–T14)

Only mild clinical signs were observed during the challenge phase. This outcome is consistent with the generally mild nature of CPiV infections, which often manifests as subclinical or mild respiratory disease in both natural and experimental settings.

In the control group, two out of twelve dogs exhibited transient respiratory signs: one developed a paroxysmal cough with retching on T8, and another presented mild purulent nasal discharge on T13. Among vaccinated dogs, only one showed a single episode of sneezing on T2.

No significant differences in body weight evolution were observed between the vaccinated and control groups during the challenge phase. Following inoculation with the virulent CPiV, dogs in both groups showed a similar body weight evolution. The mean body weight of the vaccinated and the control group ranged from 4.30 kg to 5.30 kg and from 3.92 kg to 5.01 kg, accordingly. In addition, all dogs maintained normal rectal temperatures (37.5–39.5 °C) with no significant differences between the control and vaccinated group (control group mean temperature: 38.3 °C to 39 °C; vaccinated group mean temperature: 38.3–39.1 °C).

#### 3.1.2. Seroconversion After Challenge

All vaccinated dogs demonstrated either a strong seroconversion 14 days post-challenge (T14), with seroneutralizing antibody titers ranging from 10^1.65^ to 10^1.96^. Similarly, control dogs seroconverted post-challenge, with titers ranging from 10^1.20^ to 10^1.96^, except for one dog, which did not exhibit a humoral response.

#### 3.1.3. CPiV Component Efficacy Assessment: Nasal Excretion

Nasal CPiV shedding was quantified by RT-qPCR from nasal swabs collected daily from T2 to T14. For calculation and statistical analysis, values below the limit of detection (LOD) were considered nil and attributed a value of 0.

Nasal CPiV shedding of both groups increased rapidly, exceeding 5 log_10_ copies/mL in all dogs by T2, except for two control dogs that exhibited delayed and lower viral shedding until T5, suggesting inconsistent inoculation. Both groups exhibited high and similar mean viral excretion levels until T7, peaking near T3 (approximately 8.0 Log_10_ units), as demonstrated in Figure 1.

From T7 onwards, the viral load of the vaccinated group (blue line) showed a significantly faster and steeper decline in mean viral excretion compared to the control group (red line), indicating a vaccine-associated reduction in viral shedding.

### 3.2. Experiment B (Serology Study)

#### 3.2.1. Clinical Examination: Clinical Signs, Body Weight and Rectal Temperature

Throughout the vaccination phase (D0 to D42), weekly clinical monitoring confirmed all dogs remained in good general health. They exhibited normal growth and maintained normal rectal temperatures (37.5–39.5 °C), with no clinical signs or abnormalities observed.

#### 3.2.2. CDV, CAV, and CPV Component Efficacy Assessment: Serological Responses to DHP Components

All puppies were confirmed seronegative against the vaccine components prior to immunization. Seroneutralizing antibody titers against canine distemper virus (CDV), canine adenovirus type 1 (CAV-1), canine adenovirus type 2 (CAV-2), canine parvovirus type 2 (CPV-2), canine parvovirus type 2c (CPV-2c), were assessed. Individual serological data are presented in the respective graphs in Figure 2. The individual kinetics of seroneutralizing antibody titers against four components of the Canigen^TM^ DHPPi were measured at five time points (D0, D14, D28, D35, and D42) during the vaccination phase. Titers are expressed as Log_10_ (IU/mL)
Seroneutralizing Antibody Responses against CDV

All puppies demonstrated seroconversion two weeks after the first vaccination, with seroneutralizing antibody titers ranging from 10^0.82^ to 10^1.87^. A booster effect was observed following the second vaccine administration, with titers further increasing. By four weeks post-second injection, antibody titers ranged between 10^1.05^ and 10^2.80^.
Seroneutralizing Antibody Responses against CAV-1 and CAV-2

Seroconversion was observed in all seronegative puppies at two weeks post-vaccination. By D14, antibody titers ranged between 10^1.17^ and 10^2.45^ for CAV-1 and between 10^1.98^ and 10^3.85^ for CAV-2. A secondary immune response was noted following the second vaccine injection, with titers continuing to rise. By four weeks after the second vaccination, titers ranged between 10^1.63^ and 10^3.15^ for CAV-1 and between 10^2.33^ and 10^3.38^ for CAV-2.
Seroneutralizing Antibody Responses against CPV-2 and CPV-2c

Seroconversion was observed in all initially seronegative puppies two weeks after the first vaccine dose. At D14, titers ranged between 10^3.73^ and 10^4.42^ for CPV-2 and between 10^3.62^ and 10^4.19^ for CPV-2c. The second vaccine dose resulted in an augmented immune response. By four weeks post-booster, antibody titers ranged between 10^4.19^ and ≥10^4.54^ for CPV-2 and between 10^3.73^ and 10^4.07^ for CPV-2c.

## 4. Discussion

The aim of both studies was to assess the efficacy of the Canigen^TM^ DHPPi vaccine puppies from six weeks of age, with a focus on both serological responses to CDV, CAV-1, CAV-2, CPV-2 and CPV-2c, and the clinical and viral shedding outcomes following a CPiV challenge. The findings confirm that the vaccine effectively induces robust seroconversion and confers protection against the targeted viral components.

### 4.1. Experiment A

Throughout the vaccination period, all puppies remained in good health, exhibiting normal body weight gain and rectal temperatures within the physiological range (37.5 °C to 39.5 °C). No vaccine-related adverse events were recorded, and the transient mild clinical signs observed in a few animals upon arrival at the research facility were attributed to environmental changes rather than immunization.

The CPiV challenge phase provided additional insights into the vaccine’s protective effects. Vaccinated puppies exhibited a significant reduction in viral shedding (time series, *p* < 0.04) compared to the control group from day 7 post-challenge (T7) until the end of the trial. A more rapid decline in viral load was observed in vaccinated animals, indicating a vaccine-associated decrease in viral replication and transmissibility. Clinically, signs of respiratory disease were mild and transient in both vaccinated and unvaccinated groups, with no severe manifestations recorded. This outcome is consistent with previous studies that have highlighted the challenges of achieving full protective immunity against CPiV with combination vaccines.

### 4.2. Experiment B

All puppies were confirmed seronegative against the vaccine components prior to immunization. Following the first vaccination, all animals exhibited seroneutralizing antibody titers exceeding the established protective thresholds for CDV, CAV-1, CAV-2 (≥10^0.82^), and CPV-2, CPV-2c (≥10^1.8^). The second vaccine dose resulted in a significant booster effect, with sustained high titers observed up to four weeks post-vaccination. For CDV, CAV, CPV there was a strong association between protection and the presence of virus-seroneutralizing antibodies [2,3]. These results align with previous research demonstrating that modified-live virus vaccines elicit rapid and robust immune responses in dogs [3]. Importantly, no significant variability in seroconversion was observed among individuals, suggesting a consistent immunogenic response across the study population.

It is important to acknowledge certain limitations of the studies. The serology study was designed as a single-cohort, non-blinded trial. While blinding is generally preferred to mitigate investigator bias, the risk of such bias influencing the primary results in this experiment was minimal. The seroneutralizing antibody titer measurement relies on objective, quantitative laboratory assays (Spearman–Kärber method) measured against predetermined numerical endpoints, minimizing the influence of subjective interpretation. The relatively small sample size of the studies fulfills the European Monograph criteria on vaccine efficacy [4,9,10,11].

## 5. Conclusions

The results of the studies provide evidence for the efficacy of the Canigen^TM^ DHPPi vaccine in young puppies from 6 weeks of age. Canigen^TM^ DHPPi vaccination induced serological responses against CDV, CAV-1, CAV-2, CPV-2 and CPV-2c and as soon as two weeks after the first injection, all puppies presented seroneutralizing antibody titers above the protective thresholds for these components. Furthermore, the vaccine provided protection against CPiV, as demonstrated by the challenge study. The viral load declined more rapidly and was significantly lower in vaccinated dogs compared to controls from day 7 post-challenge (T7) onwards, indicating a demonstrable reduction in viral shedding. The vaccine was well-tolerated, with all puppies exhibiting normal growth and with no significant adverse events observed during the vaccination phase. 

The study results clearly evidence the benefit of initiating the primary vaccination course with the Canigen^TM^ DHPPi vaccine in dogs from 6 weeks of age, preceding the standard, approved vaccine scheme.

The demonstrated reduction in viral excretion and the protective seroconversion on animals in this age group supports a positive benefit–risk ratio for adding this early administration. This strategy is particularly valuable in scenarios involving susceptible puppies—specifically those whose Maternal Derived Antibodies (MDA) are insufficient or wane early—and is especially critical for shelter dogs due to the higher risk of early exposure to infectious diseases and transmission.

It is important to note that the objective of this early vaccination is to initiate an immune response in susceptible puppies. Complete efficacy against CDV, CAV, CPV and CPiV, remains reliant on the complete primary vaccination schedule, as approved in the SPC of Canigen^TM^ DHPPi. Therefore, this early vaccination from 6 weeks of age must be followed by the usual vaccination scheme to provide full, long-lasting immunity until the annual revaccination.

## Figures and Tables

**Figure 1 viruses-17-01607-f001:**
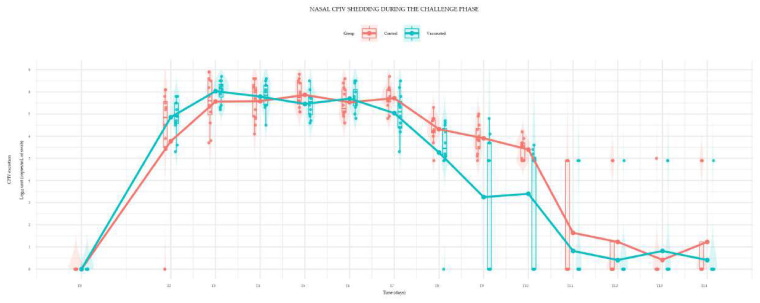
Mean Canine Parainfluenza Virus (CPiV) Nasal Shedding in Vaccinated and Control Puppies Following Virulent Challenge (Log_10_ copies/mL of swabs).

**Figure 2 viruses-17-01607-f002:**
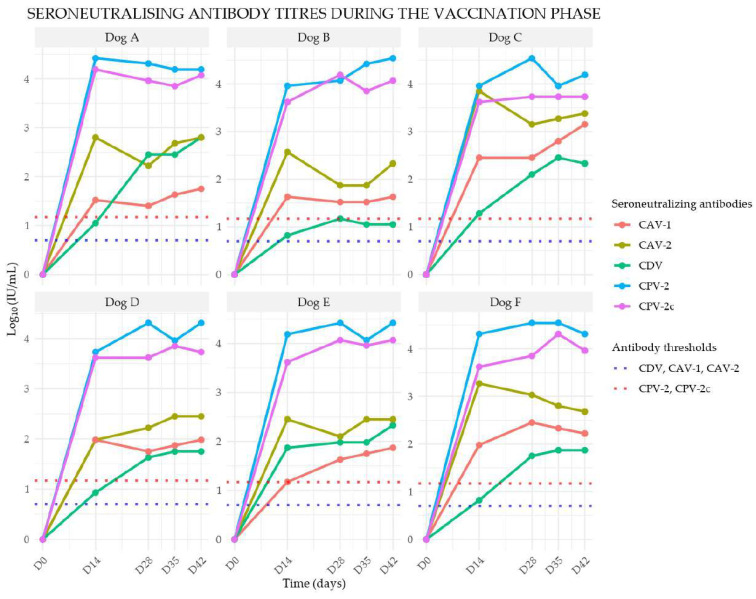
Individual seroneutralizing antibody titer profiles against the three components (CDV, CAV, CPV) of the Canigen^TM^ DHPPi vaccine in six initially seronegative puppies (Dogs A–F) following vaccination.

## Data Availability

The data presented in this study are available on request from the corresponding author. The data are not publicly available due to privacy.

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
