# Peer review of "Efficacy of a Modified-Live Virus Combination Vaccine (CDV, CAV, CPV, CPiV), CanigenTM DHPPi, in Puppies Vaccinated at Six Weeks of Age"

_viruses, 2025, doi:10.3390/v17121607_

Round 1
Reviewer 1 Report
Comments and Suggestions for Authors
The manuscript by Loukeri et al. was easily understood as an unblinded vaccine efficacy study of a modified, live-virus vaccine cocktail against canine distemper, adeno-, parvo- and parainfluenza viruses, marketed as Canigen DHPPi. The main goal of the study was to support the earlier administration of the vaccine at 6 weeks of age to provide earlier protection instead of at 8 to 10 weeks of age as is current practice to avoid interference with maternally-derived antibodies. To evaluate immunogencity to CDV, CAV, and CPV-2, 6 six-week-old Specific Pathogen Free puppies were serologically screened. To evaluate protective efficacy against CPiV, a challenge study was conducted using 24 six-week-old puppies, with 12 receiving the vaccine and 12 receiving diluent only. Results state that antibody responses were robust, and animals maintained good health with no adverse reactions due to vaccination. Of note, this early vaccination would not replace any of the standard course of vaccinations against CDV, CAV, CPV and CPiV.
The following are some revisions, which are either suggested or required to be addressed prior to publication, as noted:
The Keywords "distemper", "hepatitis", "parvovirus", "parainfluenza", "challenge" and "seroconversion" are suggested for inclusion. (Line 42)
Figures must be revised in order to be publishable. Unfortunately, the figures could not be fully evaluated, as graphs, axes, titles and legends are truncated or missing. Both when viewing online and when dowloaded, the figures are not properly displayed.
Figure descriptions are very brief and do not assist the reader in understanding the data, especially considering the absence of figure legends in its current form (Lines 220 and 253).
The in-text citation of Figure 1 was not found in section 3.1.3 (Line 216).
Under Materials and Methods section 2.1, please include more information about the puppies used for each study. Generally, some representation of breed information is desired, whether purebred or mixed, representative breeds, source, housing, etc. (Line 79)
Please include the definition for the abbreviation DHPPi. (Line 334)
Author Response
Comment 1: The Keywords "distemper", "hepatitis", "parvovirus", "parainfluenza", "challenge" and "seroconversion" are suggested for inclusion.
Response 1: The suggested keywords were added in the relative section: Keywords/Line 43-44
Comment 2: Figures must be revised in order to be publishable. Unfortunately, the figures could not be fully evaluated, as graphs, axes, titles and legends are truncated or missing. Both when viewing online and when downloaded, the figures are not properly displayed.
Reply 2: Both Figures are revised (quality, legends, title, description in the text)
Comment 3: Figure descriptions are very brief and do not assist the reader in understanding the data, especially considering the absence of figure legends in its current form (Lines 220 and 253).
Reply 3: The description of Figure 1 is added in section 3.1.3/Line 232-239. The description of Figure 2 is added in section 3.2.1/Line 254-276
Comment 4:The in-text citation of Figure 1 was not found in section 3.1.3 (Line 216).
Reply 4: In-text citation of Figure 1 is added in section 3.1.1/Line 236
Comment 5:Under Materials and Methods section 2.1, please include more information about the puppies used for each study. Generally, some representation of breed information is desired, whether purebred or mixed, representative breeds, source, housing, etc. (Line 79)
Reply 5: Additional information for the puppies is now included in section 2.1.1./ Line 88 to 92 and section 2.1.2 /Line 97-101
Comment 6:Please include the definition for the abbreviation DHPPi. (Line 334)
Reply 6: It is now included in the abbreviations section
Reviewer 2 Report
Comments and Suggestions for Authors
This study evaluated the efficacy of a commercial multivalent vaccine in puppies following immunization. However, there are several issues with the experimental design of the paper. First, in Experiment A, the puppies used were not tested for the presence of various pathogens before immunization. The study should have pre-determined the source of the puppies, their pathogen infection status, and corresponding antibody levels. In Experiment A, why was Canine Parainfluenza Virus (CPiV) selected for the challenge test? What is the basis for the chosen challenge dose? Furthermore, in the study, there appeared to be no significant difference in clinical manifestations between the control group and the challenge group in Experiment A. It is recommended to score and quantify indicators such as clinical symptoms, body weight, and body temperature. In Experiment B, how were neutralizing antibodies detected? What is the quantitative basis for the levels of neutralizing antibodies against Canine Distemper Virus (CDV), Canine Parvovirus (CPV), and Canine Adenovirus (CAV)? It is difficult to conclude that the neutralizing antibody titers of all puppies against these components exceeded the protective thresholds solely based on neutralizing antibodies without standard references. Additionally, the information presented in Figures 1 and 2 is incomplete.
Author Response
Comment 1: In Experiment A, the puppies used were not tested for the presence of various pathogens before immunization. The study should have pre-determined the source of the puppies, their pathogen infection status, and corresponding antibody levels
Reply 1: Additional information for the status of the puppies prior to vaccination is added in section 2.1.1/Line 88 to 92
Comment 2: In Experiment A, why was Canine Parainfluenza Virus (CPiV) selected for the challenge test?
Reply 2: Antibody level (seroneutralization titer) is not reliably indicative of protection against Canine Parainfluenza (CPiV). Efficacy relies heavily on local, mucosal immunity in the respiratory tract (IgA antibodies, cell-mediated immunity), which is poorly correlated with systemic (serum) antibody titers. As demonstrated in literature, in the evaluation of parenteral modified-live CPIV vaccine, there was no direct correlation between the amount of serum antibody level and duration of shedding. The only way to confirm true protection is to directly expose vaccinated animals to a virulent form of the virus (a challenge study) and observe the prevention or reduction of clinical signs and viral shedding. [Ellis, J.A. ; Krakowka G.S. A review of canine parainfluenza virus infection in dogs. J Am Vet Med Assoc. 2012, 240, 273-84. doi:10.2460/javma.240.3.273.]
The following phrase is added in the manuscript, section 2.1.1./ Line 83 to 88.
- As demonstrated in literature, the only way to confirm true protection is to directly expose vaccinated animals to a virulent form of the virus (a challenge study) and observe the prevention or reduction of clinical signs and viral shedding. Antibody level (seroneutralization titer) is not reliably indicative of protection against Canine Parainfluenza, given that efficacy relies heavily on local, mucosal immunity in the respiratory tract which is poorly correlated with serum antibody titers
Comment 3: What is the basis for the chosen challenge dose?
Reply 3: The challenge dose was determined through dose titration studies conducted by Virbac's Clinical Development Unit. These studies were designed to confirm the virulence and consistency of the Canine Parainfluenza Virus (CPiV) challenge strain in the target host.
Study Design: Dogs aged 12 to 13 weeks were inoculated with CPiV doses ranging from 104 CCID50/dog up to 4.105 CCID50/dog. The inoculum was prepared in Minimum Essential Medium (MEM) and administered as a 2 x 1 ml of suspension via both the intra-tracheal and intranasal routes2.
Virulence Assessment: Pathogenicity was evaluated over 14 days post-inoculation based on the severity of respiratory clinical signs and the duration of viral excretion. Viral excretion was monitored via nasal swabbing followed by CPiV re-isolation on cell culture.
All challenged dogs across the entire dose range (from the lowest dose of 104 CCID50/dog to the highest dose to 4.105 CCID50/dog demonstrated viral excretion throughout the 14-day challenge period and respiratory clinical signs. These results established that the lower dose of 104 CCID50/dog was sufficient to consistently induce both viral shedding and clinical signs, thus validating this dose as the minimal effective challenge dose for use in subsequent efficacy studies, in compliance with regulatory standards for viral challenge strains. The dose used in the efficacy trial was, therefore, 104 CCID50 per dog.
The following phrase has been added to the manuscript, in section 2.3.1/Line 122 to 125: The challenge dose was validated in dose titration studies demonstrating that 4.105 CCID50/dog consistently induced both viral shedding and respiratory clinical signs in 12–13 week old dogs. This dose was therefore selected as the minimum effective challenge dose.
Comment 4: Furthermore, in the study, there appeared to be no significant difference in clinical manifestations between the control group and the challenge group in Experiment A. It is recommended to score and quantify indicators such as clinical symptoms, body weight, and body temperature.
Reply 4: Additional information on this part is added in section 3.1.1./Line 215 to 222
Comment 5: In Experiment B, how were neutralizing antibodies detected? What is the quantitative basis for the levels of neutralizing antibodies against Canine Distemper Virus (CDV), Canine Parvovirus (CPV), and Canine Adenovirus (CAV)? It is difficult to conclude that the neutralizing antibody titers of all puppies against these components exceeded the protective thresholds solely based on neutralizing antibodies without standard references.
Reply 5: The neutralizing antibodies were detected by immunofluorescence (IF) using a virus specific antibody. The titre is calculated according to the Spearman and Kärber method to determine the end point dilution, at which the cytopathogenic effect induced by the virus is reduced by half.
Historical studies supported serology as an indicator of efficacy. Several studies have been performed to demonstrate a correlation between sero neutralising antibodies and protection against challenge (clinical signs/mortality) for CPV, CDV and CAV-2 vaccine components.
Information on the detection of the antibodies has been added to section 2.4.1.3/Line 171 to 178
Comment 6: Additionally, the information presented in Figures 1 and 2 is incomplete.
Reply 6:
- Both Figures are revised (quality, legends, title, description in the text)
- The description of Figure 1 is added in section 3.1.3/Line 232-239
- The description of Figure 2 is added in section 3.2.1/Line 254-276
Reviewer 3 Report
Comments and Suggestions for Authors
Dear authors, the comments are attached.

Author Response
Comment 1: Both experiments use small sample sizes, which is common in regulatory safety/efficacy trials, but some justification is needed. For example: experiment A uses n=12 per group, which is acceptable but should include a brief statement on how this number aligns with EP monographs or prior power expectations. Experiment B uses n=6, making between animal variability more impactful. Add one or two sentences acknowledging limitations related to sample size and how the design still fulfils the regulatory efficacy criteria.
Reply 1: The acknowledgment of the limitations both of sample size and the non-blinded serololigical evaluation is added in the Discussion section/ Line 315 to 322
Comment 2: The serology study is explicitly non-blinded, which introduces potential bias when interpreting clinical and laboratory endpoints. Explicitly discuss this limitation in the Methods or Discussion and justify why the risk of bias is minimal (e.g., lab-based endpoints, objective neutralization assay).
Reply 2: The acknowledgment of the limitations both of sample size and the non-blinded serololigical evaluation is added in the Discussion section/ Line 315 to 322
Comment 3: The text states positivity thresholds (e.g., 10^0.7, 10^1.17), but how these relate to clinically protective levels is not entirely clear for all pathogens. Add a brief justification or citation explaining how these thresholds correspond to protection (e.g., WSAVA guidelines, previous efficacy studies).
Reply 3: The following justification is added in the manuscript in section 2.4.1.3/Line 174 to 178:
The designated positivity thresholds are based on internal studies validating the Correlation of Protection between seroneutralizing antibodies and protection against clinical disease and mortality following a virulent challenge for the CDV, CPV and CAV-2 vaccine components. The study reports were submitted to European regulatory bodies such as the European Medicines Agency (EMA).
The following justification is added in the manuscript in section 2.4.1.3/Line 174 to 178
Comment 4: Figure 1 conveys trends but lacks confidence intervals or individual animal traces, which would provide useful granularity. If possible, include error bars or shaded confidence regions. Clarify in figure legend whether data points represent means of log10 values ± SEM or ± SD.
Reply 4: Individual animal traces, Box-plots and Shaded confidence intervals are added in Figure 1. The data points show mean log 10 values. This information is added in the legend.
Comment 5: The manuscript repeatedly emphasizes that respiratory signs were mild in both groups. Given the nature of CPiV challenges, this is expected. Add one sentence explaining that mild clinical disease is typical for CPiV infection, thereby contextualizing the findings.
Reply 5: This clarification is added on section 3.1.1./Line 208 to 210:
Only mild clinical signs were observed during the challenge phase. This outcome is consistent with the generally mild nature of CPiV infections, which often manifests as subclinical or mild respiratory disease in both natural and experimental settings.
Comment 6: The use of mixed models is appropriate, but please state the covariance structure chosen (e.g., AR(1), unstructured).
Reply 6: In Figure 2 the lines do not represent mixed models. They represent the seroneutralizing trends of each vaccine component (CAV-1, CAV-2, CDV, CPV-2 and CPV-2c) for each individual dog. It is just simple representation of the measurement.
Comment 7: Ensure consistent use of: CPV-2 vs CPV-2C; Log10 formatting (sometimes 10^1.17 is written differently); Seroneutralizing antibody titers” vs “neutralizing titers”.
Reply 7:
- We agree that some modifications for consistency were necessary and the manuscript was corrected in section 2/ 2.3.3 / 2.4.1.3 / 4.2.
- The abbreviation CPV is used when the text refers in general to Canine Parvovirus, the abbreviation CPV-2 and CPV-2c is used when it is necessary to specify that serological protection was demonstrated for both strains of parvovirus.
- We could not identify the specific parts were the log10 formatting was different but we reviewed again all the formatting. We hope that now is consistent.
- The relevant modification are done in the manuscript and the term “seroneutralizing” is maintained.
Comment 8: A few sentences are duplicated in the results (e.g., clinical description around D21–D28). Removing redundancy will improve flow.
Reply 8: Yes, we have identified this and the duplicates have been removed.
Comment 9: Consider consolidating ethics statements into a single paragraph for clarity.
Reply 9: The ethics statement has been modified accordingly.